# A Peptide-Based Nanocarrier for an Enhanced Delivery and Targeting of Flurbiprofen into the Brain for the Treatment of Alzheimer’s Disease: An In Vitro Study

**DOI:** 10.3390/nano10081590

**Published:** 2020-08-13

**Authors:** Shafq Al-azzawi, Dhafir Masheta, Anna Guildford, Gary Phillips, Matteo Santin

**Affiliations:** 1Centre for Regenerative Medicine and Devices, School of Pharmacy and Bimolecular Sciences, University of Brighton, Brighton BN2 4GJ, UK; phar.shafaq.kadhim@uobabylon.edu.iq (S.A.-a.); phar.dhafir.qahtan@uobabylon.edu.iq (D.M.); a.guildford@tissueclick.com (A.G.); gphillips83@btinternet.com (G.P.); 2College of Pharmacy, University of Babylon, Ministry of Higher Education and Scientific Research, Hilla 51002, Iraq; 3Tissue Click Ltd., Brighton BN2 6SJ, UK

**Keywords:** Alzheimer’s disease, neurodegenerative diseases, blood–brain barrier, ApoE-peptide, drug delivery system, flurbiprofen

## Abstract

Alzheimer’s disease (AD) is an age-related disease caused by abnormal accumulation of amyloid-β in the brain leading to progressive tissue degeneration. Flurbiprofen (FP), a drug used to mitigate the disease progression, has low efficacy due to its limited permeability across the blood–brain barrier (BBB). In a previous work, FP was coupled at the uppermost branching of an ε-lysine-based branched carrier, its root presenting a phenylalanine moiety able to increase the hydrophobicity of the complex and enhance the transport across the BBB by adsorptive-mediated transcytosis (AMT). The present study explores a different molecular design of the FP-peptide delivery system, whereby its root presents an ApoE-mimicking peptide, a targeting ligand that could enhance transport across the BBB by receptor-mediated transcytosis (RMT). The functionalised complex was synthesised using a solid-phase peptide synthesis and characterised by mass spectrometry and FTIR. Cytotoxicity and permeability of this complex across an in vitro BBB model were analysed. Moreover, its activity and degradation to release the drug were investigated. The results revealed successful synthesis and grafting of FP molecules at the uppermost molecular branches of the lysine terminal without observed cytotoxicity. When covalently linked to the nanocarrier, FP was still active on target cells, albeit with a reduced activity, and was released as a free drug upon hydrolysis in a lysosome-mimicking medium. Noticeably, this work shows the high efficiency of RMT-driven FP delivery over delivery systems relying on AMT.

## 1. Introduction

Alzheimer’s disease (AD) is an uncurable age-related neurodegenerative disease (ND) leading to a progressive loss of memory and cognition for which the incidence is exacerbated in an ageing population [1,2,3]. The main pathological hallmarks of the disease are abnormal aggregates of toxic extracellular deposits of beta-amyloid (Aβ) peptides and tau amyloid proteins in the brain [4]. Aβ species are generated from the amyloid precursor protein (APP) via sequential proteolytic cleavages mediated by γ-secretase enzyme. The fragments of APP are made up of different Aβ isoforms depending on the number of amino acids [5]. The most prevalent isoforms of Aβ are Aβ40 and Aβ42 which, in AD, form dense clumps of deposits surrounding neurons and giving rise to senile plaques (SP) of ND [6]. In healthy individuals, the Aβ peptide level remains at a steady state and does not lead to the generation of amyloid plaques due to metabolism of the peptides [7].

Nonsteroidal anti-inflammatory drugs (NSAIDs) such as Flurbiprofen (FP) have been considered in the treatment of AD because of their activity as γ-secretase modulators (GSMs) [8]. FP is able to reduce Aβ42 levels by selectively controlling the γ-secretase activity without remarkable impediment to the Notch signalling or other pathways of APP processing [9]. This selective action mode brings significant clinical advantages when compared to nonselective γ-secretase inhibitors, which can affect the metabolism of total amyloid proteins that regulate many neuronal and synaptic functions [10]. In addition, FP is an FDA-approved drug, commercially used for pain relief and anti-inflammatory action [11]. The drug is well-absorbed within a 4-h biological half-life and is fully excreted from the body within 24 h [12]. However, FP has a low BBB penetration and the reached concentration at the target brain tissue is below the required therapeutic concentration (150−250 μM) reported to exert a significant pharmacological effect on the γ-secretase activity [13,14].

Various approaches have been developed to enhance the BBB permeability towards FP, including invasive techniques, chemical modifications, and enhanced permeability of the endothelial cells of the BBB via endogenous transcytosis [15]. In particular, adsorptive mediated transcytosis (AMT) and receptor-mediated transcytosis (RMT) have been considered as the most favourable ones to enhance drug transport without damaging the BBB cells [16]. AMT allows essential circulating hydrophobic molecules to enter the brain via charge interaction between positively charged molecules and negative charges on the cell membrane [17]. Our previous work has demonstrated the ability of a peptide-based nanocarrier to enhance FP transport across brain endothelial cells by AMT when a hydrophobic phenylalanine moiety was integrated with a lysine and where FP molecules were covalently coupled to the two amino terminals of the latter [18].

However, RMT, through a specific receptor of the BBB endothelial cells, has been shown to provide more selective and active targeting [19]. Different macromolecules and drug-laden nanocarriers have been shown to be efficiently transported by this approach [16,20,21], particularly exploiting the enhanced expression of these specific receptors by the endothelial cells of the BBB [21,22,23].

There are various potential endogenous receptor systems that have been exploited for a range of applications including the insulin receptor, but this is considered the riskiest approach as its mechanism interferes with glucose homeostasis [15,24]. An alternative receptor target is the transferrin (Tf) receptor (TfR), which mediates the transport of TF-bound iron that has been exploited to transport drugs across the BBB either by using Tf as a ligand or by using a specific antibody (such as OX26) [25]. However, Tf’s relatively high molecular weight and limitations in its scaled-up synthesis hinder its use in pharmaceutical formulations [15]. The diphtheria toxin receptor is another transmembrane receptor with no particular endogenous ligand, but it is not widely used because of its toxicity [15,26].

Currently, the most widely targeted receptor to deliver drugs or genes to the brain is the low-density lipoprotein receptor (LDLr) for the respective Proteins 1 and 2 (LRP1r and LRP2r), as they are upregulated on the BBB endothelia when compared to other types of tissue cells [27,28,29,30]. Extensive studies have been performed on the interaction of this receptor with ApoE when expressed by brain endothelia, and it has been demonstrated that nanoparticles (NPs) decorated with ApoE have a higher BBB penetration through LDLr-mediated transcytosis [31]. In particular, a specific amino acid sequence of ApoE has been identified as promoting the RMT thus becoming a suitable candidate for BBB RMT nanocarriers [32,33,34,35].

This binding sequence of the ApoE has been investigated in various studies [36,37], and it was initially found that the amino acid sequence (141-155) can indeed be recognised by the LDLr [38]. A shorter sequence (i.e., 141-150, -LRKLRKRLLR-) has been shown to induce endocytosis via LDLr-engagement using primary neuronal cultures of brain tissue [39]. The endocytosis process of this sequence has been further investigated via electron and fluorescence microscopy and found to be able to interact with the LDLr [33]. The decoration of liposomes with this peptide sequence (141-150) has been efficiently shown to be taken up by the brain endothelium of rats [33]. Different carrier systems, such as nanoliposomes (NLs), to deliver drugs were successfully synthesised and functionalised with the ApoE-derived residue and evidently enhanced the brain uptake of antioxidant drugs and curcumin [32,34].

Our previous work has demonstrated that this sequence can enhance the uptake of a peptide-based nanocarrier by brain endothelial cells [40]. The study showed that the ApoE-mimicking sequence—LRKLRKRLLR—could be modified by lysine monomers exposing amino terminal branches able to enhance drug coupling and targeting brain endothelial cells via RMT.

Capitalising on these previous studies, the present work shows the optimised method of designing a G0 ApoE-derived peptide (AEP) delivery system to which two FP molecules were coupled at the two lysine amino terminals to improve the bioavailability of the drug at brain tissue. The work showed that this drug delivery system has no cytotoxicity and could exert FP inhibition on the γ-secretase activity of neuroglial cells, either when the drug was still coupled to the peptide carrier or when released through hydrolysis in a medium mimicking that of the lysosomes. Noticeably, the transcytosis of FP across a brain endothelium model was enhanced in comparison to that observed for both the free drug and an AMT-driven FP peptidic nanocarrier [18,40]. The aim of this study is, therefore, to design a biocompatible and biodegradable drug delivery system able to carry a higher FP payload across the BBB and release it at the target (brain tissue), with prospective optimal bioavailability and a concentration that should minimise side effects seen with current high-dose therapy in the treatment of AD.

## 2. Materials and Methods

### 2.1. Design and Synthesis of the RMT-Driven (AEP-Functionalised)-FP Delivery System

The RMT-driven-FP delivery system was designed by modifying the ApoE-mimicking sequence, H-LRKLRKRLLR-OH, with the addition of a lysine (K) at its amino terminal and coupling two FP molecules to the two exposed amino groups of the lysine. The complex AEP-lysine-FP delivery system (AEP-K-FP) (chemical formula: C_96_H_152_ F_2_N_26_O_14_, molecular weight (MW): 1932 Da (Figure 1) was obtained by a solid phase peptide synthesis (SPPS) method, where the yield of the reaction was enhanced by the aid of microwave synthesiser Biotage Initiator (Hengoed, UK) [41]. All amino acids, Rink amide, and Tenta gel used were obtained from Merck Ltd. (Southampton, UK) (purity 98%), and other solvents of high-performance liquid chromatography (HPLC) grade were purchased from Fisher Scientific (Loughborough UK). The assembly started on a Tenta gel NH_2_ resin (0.5 g) using 0.4 mmol of phenylalanine (Fmoc-Phe-OH) and a sequence of Fmoc-Leu-OH, Fmoc-Arg-OH, Fmoc-Lys(Boc)-OH, Fmoc-Leu-OH, Fmoc-Arg-OH, Fmoc-Lys(Boc)-OH, Fmoc-Arg-OH, Fmoc-Leu-OH, Fmoc-Leu-OH, Fmoc-Arg-OH, consecutively followed by the coupling of lysine (Fmoc-Lys(Fmoc)-OH) and FP (Sigma Aldrich, Gillingham, UK) [40].

In particular, the synthesis process included a series of coupling and deprotection for each added amino acid. For each amino acid coupling, 0.4 mmol N-[1H-benzotriazol-1-yl)-(dimetylamino)methylene]-N-methylmethanaminium hexafluorophosphate N-oxide (HBTU), 140 µL N,N–diisopropylethylamine (DIPEA), and 3 mL N, N–dimethylformamide (DMF) were added and sonicated for solubilisation. However, the deprotection that removes the Fmoc group involved the addition of 3 mL solution of 20% *v/v* piperidine/DMF [42]. Furthermore, a capping step was added during the synthesis to remove any unreacted amino groups or separated truncated sequences that might be formed after the coupling steps, so the loading of the resin could also be reduced. The capping is achieved by treating the peptide when still on the resin, after coupling, with a high excess (approximately 50-fold molar) of a reactive derivative of acid and a base [43]. In this work, when the coupling solution was completed, the capping step was performed according to a previously published method [44]. When all amino acids were coupled, the two FP molecules were covalently grafted to the lysine amino terminal using the same method used for deprotected amino acids. After that, the synthesised molecule was cleaved from the resin using a cocktail solution of 95% trifluoroacetic acid, 2.5% triisopropylsilane, and 2.5% (*v/v*) deionised water. The cleaved mixture was then filtered, collected, and purified from any undesired by-products by Zeba spin desalting columns (Fisher Scientific, Loughborough, UK). The final pure product was stored dry at −20 °C until used for all the subsequent experiments.

### 2.2. Characterisation of the AEP-Functionalised FP-Delivery System

#### 2.2.1. Mass Spectrometry

The synthesised delivery system was characterised by electrospray/ionisation time-of-flight (ESI-TOF MS) (Bruker Daltonics, Coventry, UK) at high voltage (4 kV). In this mode, sample mass (m/z) gives rise to multiple-charged ions associated with charges (n) as (MW + nH)/n, (H is represented as mass of proton = 1.008Da).

#### 2.2.2. Fourier Transform Infra-Red (FTIR)

FTIR spectroscopy (Perkin Elmer Spectrum 65, Llantrisant, UK) was used to investigate the structural and functional group changes in the synthesised complex using a few milligrams of the sample via 32 scans in the range 550–4000 cm^−1^.

### 2.3. Preparation of Cultured Cells

#### 2.3.1. bEnd.3 Cells

Immortalised brain endothelial cells, bEnd.3 (passages 15–20), were cultured, according to the (manufacture ATCC) product sheet, in a Dulbecco’s modified eagle’s medium (DMEM) high glucose medium with L-pyruvate, containing 10% (*v/v*) foetal bovine serum (FBS) and 1% (*v/v*) of 500 U/mL Penicillin/Streptomycin (Gibco, Gaithersburg, MD, USA). Cells were seeded at a density of 5 × 10^4^ cells per cm^2^ in 24-well plates, which were incubated at 37 °C and 5% CO_2_ with replacement of the culture media every 3 days.

#### 2.3.2. Human Umbilical Vein Endothelial Cells

The synthesised AEP-K-FP was also tested on human umbilical vein endothelial cells (HUVECs) (ATCC- Manassas, VA, USA) (passages 35 to 39). HUVECs were routinely cultured in an endothelial basal medium (F-12K) containing 0.05 mg/mL of endothelial cell growth supplement (ECGS), 0.1 mg/mL of heparin (Gibco Gaithersburg, MD, USA) and adjusted to a final concentration of 10% *v/v* of FBS (according to the manufacturer’s recommendations).

#### 2.3.3. Preparation of C6 Glial Cells

C6 glial cells (ATCC-) (passages less than 20) were seeded using the complete growth medium, which was made according to the ATCC product sheet by addition of FBS to a final concentration of 2.5% *v/v* and horse serum (Gibco Gaithersburg, MD, USA) to a final concentration of 15% *v/v* to F-12K medium. Experiments were performed when cells reached confluence.

### 2.4. Cytotoxicity Assays

Experiments were performed when cells reached confluency and a range of concentrations (25 to 400 µM) of AEP-K-FP were used in each experiment. The 3-(4,5-dimethylthiazol-2-yl)-2,5-diphenyltetrozolium bromide (MTT) assay (Sigma Aldrich, Gillingham, UK) was used to measure cell viability [45]. After 24 and 48 h of treatment exposure, the absorbance was measured at a wavelength of 540 nm in a spectrophotometer (Thermo Multiskan Ascent, Rochford, UK). Readings were expressed as a percentage of the untreated control cells.

The lactate dehydrogenase (LDH) assay provides an indication of loss of cell membrane integrity and was measured using a Promega CytoTox96^®^ nonradioactive cytotoxicity assay kit (Southampton, UK) after 24 and 48 h of treatment. Absorption was read spectrophotometrically at 492 nm and converted to a percentage of the total LDH released from the positive control (untreated cells with complete lysis).

For cell death assessment, Hoechst 33342 and propidium iodide (PI) salt (HPI) was used (Sigma-Aldrich, Gillingham, UK). These intracellular dyes stain cells in which the Hoechst 33342 binds to DNA, so it is staining the nuclei, whereas PI stains the cytoplasm, resulting in emission of light observed under a fluorescent microscopy [46]. Live cells appear blue with intact membranes on microscopic examination, and those at early and late apoptotic phases appear bright blue with shrinkage. On the other hand, necrotic or damaged cells appear red with ruptured membranes [47].

HPI dye was prepared by adding 900 µL of DMEM to a mixture solution of 50 µL of 1mg/mL Hoechst 33342 and 50 µL of 2 μg/mL PI in a glass vial. After 24 h of treatment and incubation, which were performed as previously discussed [44], an epi fluorescent microscopy (Olympus microscope equipped with a Nikon D1X camera, UK) was used to identify and count the cells with distinct morphologies. Cell counts for healthy (living), apoptotic, or necrotic cells were obtained from the same treatment in duplicate wells and from three different fields for each well. Counting of the cells was measured according to their appearance then averaged and expressed as a percent of the total cell number.

### 2.5. Examination of the Drug Delivery System Penetration Across an In Vitro BBB Model

The lyophilised powders of both AEP-K-FP and the free drug (FP) were dissolved in the FBS-free, phenol-free culture media in a concentration of 200 µM. Quantitative analysis of penetration for the free-drug and drug-attached delivery system was undertaken using HPLC (Agilent technology, Stockport, UK). The analysis was carried out using a hydrophobic C18 column (150 × 4.6 mm) at 25 °C and 20 μL of injection volume. The flow rate used was 0.6 mL/min, while the UV-detection wavelength was 248 nm, which is the λ_max_ of FP [9]. The HPLC mobile phase solvents consisted of water/acetonitrile in which the gradient of eluent was run from 75:25 to 25:75 water: acetonitrile over 20 min. The standard curve for each molecule was obtained to calculate the percentage of the amount that transported across the in vitro BBB model.

To establish an in vitro BBB model, bEnd.3 cells were seeded on microporous membrane of 12-transwell inserts with complete culture media in a routine manner. Trans-endothelial electrical resistance (TEER) between apical and basolateral chambers was measured using an Evom voltometer (Sarasota, FL, USA) to determine the day of maximum values indicating tight junction formation, as previously described [18].

In addition, paracellular permeability of this model was investigated using sucrose as a marker, which was measured by the glucose oxidase (GOD)/invertase method, where its low permeability indicated the tightness of the barrier [48]. The cells were cultured on transwell inserts (membrane of 0.4 µm pore size, Fisher Scientific, Loughborough, UK) to confluence, then treated with 1 mL solution of sucrose/phosphate buffer saline (PBS) at a concentration of 2 mg/mL. After 15 and 30 min, samples were collected from the lower chambers. In separate microcentrifuge tubes, 5 µL of sample, 85 µL distilled water (DW), and 10 µL invertase /DW solution (of 10 mg/mL concentration) (Sigma Aldrich, Gillingham, UK) were added. Control tubes that contained only PBS instead of the sample were also included.

The tubes were placed in a water bath at 55 °C for 10 min, then 200 µL of GOD reagent (Sigma Aldrich, Gillingham, UK) was added and the tubes were again placed in a water bath at 37 °C. After 15 min, the tubes were removed, and 10 µL of phenol solution was added to be incubated at room temperature for 5 min. Samples were taken in duplicate and the absorbance was read at 490 nm using a spectrophotometer. A calibration curve was constructed utilising standard sucrose solutions to determine the sucrose concentration in the samples.

### 2.6. Quantification of γ-Secretase Enzyme Activity

The investigation is based on sandwich enzyme-linked immunosorbent assay (ELISA) technology using a γ-secretase enzyme kit (Abbexa Ltd., Cambridge, UK). After confluence, C6 glial cells were treated with 200 µM FP or AEP-K-FP / F-12K solution. Wells containing untreated C6 cells were also included as a control. After 4 h of incubation, cells were washed and lysed to be centrifuged, and the supernatant was collected for assaying the γ-secretase activity immediately, according to kit instructions. Data were expressed as pg/mL, following kit instructions.

### 2.7. Drug Release

The AEP-K-FP was analysed for the hydrolysis of the amide linkage that coupled the drug molecules to the peptide carrier as well as its amino acids residues. This hydrolysis of the complex was evaluated in an acidic solution mimicking the lysosomal pH conditions that are likely to be found in an inflamed tissue such as that of AD-affected brain tissue. The acidic buffer solution was prepared using a phthalate solution, according to manufacturer’s instructions (Thomas Scientific, Swedesboro, NJ, USA), to obtain a final pH of 4.5. The AEP-K-FP was dissolved in this buffer solution at a concentration of 200 µM and incubated at static conditions for 24 h, 37 °C. Samples were collected and analysed by HPLC.

### 2.8. Statistical Analysis

Mean values and standard deviation (SD) were calculated from the number of readings (n) from each experiment. Results were statistically analysed using one-way ANOVA with Tukey’s tests. Significant difference was identified by a *P* value < 0.05.

## 3. Results

### 3.1. Characterisation of AEP-Functionalised FP-Delivery System

The mass spectra showed the successful synthesis of the AEP-K-FP with peaks corresponding to the charged form of the expected MW, 1932 Da (Figure 2). Other peaks appearing in the spectra were attributed to the ionisation of the molecule and to the solvent or machine noise, as well as to the sodium salts formed due to interaction of ions with glass vessels [49].

FTIR analysis showed the shifting of peaks caused by the formation of an amide linkage at 3200 cm^−1^ and 1640 cm^−1^ and appearance of an aromatic ring peak of FP in the FP-linked AEP-K, confirming the attachment of the drug to the peptidic nanocarrier (Figure 3).

### 3.2. Cytotoxicity

An MTT assay was carried out on confluent bEnd.3 and HUVEC cells after 24 and 48 h of treatment with increasing concentrations of the AEP-K-FP. The results showed no reduction in cell metabolic activity by giving values more than 70% in relation to the control untreated cells (Figure 4), and all values were within acceptable cytotoxic range (International-Standards 2009). The results suggest that AEP-K-FP was not toxic to either cell phenotypes, even at the highest concentration used (400 µM) they were 78.7% and 76.8% in bEnd.3 at 24 and 48 h, respectively, whereas for those with HUVEC, they were 83% and 80.2% after 24 and 48 h of treatment, respectively. In addition, the statistical analysis showed no significant differences (*P* > 0.05) between the corresponding concentrations after 24 or 48 h of exposure.

No cytotoxicity was found when LDH analysis was performed in the same range of concentration used for the MTT experiments. Values of LDH release were less than 50% in relation to the positive controls with a significant difference at *P* < 0.001. The highest values of LDH release after 24 and 48 h of treatment with AEP-K-FP were seen at a concentration of 400 µM for both cultured cells, with no significant differences (*P* > 0.05) between each other at corresponding concentrations (Figure 4).

HPI staining of brain endothelial cells following treatment with AEP-K-FP revealed no cytotoxicity consistently with the results obtained from both MTT and LDH experiments. The percentages of healthy cells at the concentration range used (100, 300, and 400 µM) were never lower than 91%. While the values of apoptotic and necrotic cells did not exceed 4.7% and 3.4%, respectively, and no significant differences (*P* > 0.05) from the control samples at any concentrations were observed (Figure 5).

### 3.3. Penetration of the Drug Delivery System Across an In Vitro BBB Model

The penetration of both the free drug and the RMT-driven drug delivery system following 1h and 4 h of incubation was evaluated by a BBB model using bEnd.3 cultured as monolayers on a transwell system, which was created and validated using TEER measurement, as reported in a previous work [18]. To verify that this cultured monolayer is devoid of paracellular transport for solutes, sucrose was used as a permeability marker. The standard curve of a series of dilutions of the sucrose solution was carried out to quantify the amount of sucrose that penetrated across the membrane, and the percentage of permeability was calculated, as shown in Table 1.

After a 15 min incubation of the cultured cells (bEnd.3 and HUVEC) in the sucrose solution on transwell inserts, only 6.2% of the initial amount passed across the bEnd.3 monolayer in comparison to 19.6% for that with HUVEC, whereas the amount did not increase more than 9.4% and 32.2%, respectively, after 30 min. The membrane alone showed free sucrose transport to give equilibrium between the two chambers. These results demonstrated the resistance of the formed bEnd.3 monolayer to paracellular solute transport.

A series of dilutions of both the free drug and the AEP-K-FP dissolved in phenol-free DMEM solution were analysed by HPLC, and the relative standard curves were obtained (Figure 6).

The percentage of the AEP-K-FP permeated across this model increased more than 6 times (16.3%) after 1 h, and 4 times (35.9%) after 4 h with significant differences (*P* < 0.001) from that of free-FP levels, which did not exceed 2.7% and 8.5% of the applied concentration after 1 h and 4 h, respectively (Figure 7). It is worth mentioning that the AEP-K-FP has 2 additional moles of drug, hence this can increase the amount of drug that is released upon dissociation of the linkage.

### 3.4. Quantification of γ-Secretase Enzyme Activity

As demonstrated in Table 2, values obtained for the amount of γ-secretase from the set of C6 control cells were 28.4 pg/mL, while treatment cells with FP significantly decreased (*P* < 0.05) the amount to 8.2 pg/mL. The analysis showed a significant decrease (*P* < 0.05) in γ-secretase for the cells that were incubated with AEP-K-FP in comparison to the control, with average values of 14.6 pg/mL. However, the decrease in the enzyme activity induced by AEP-K-FP was not as much as that observed with free drug (*P* < 0.05) (Table 2).

### 3.5. Degradation Analysis and Drug Release

To investigate the hydrolysis of the AEP-functionalised FP-delivery system and the release of the attached drug in an environment mimicking the inflamed tissue typical of AD conditions, an acidic buffer was used. Qualitative HPLC analysis of the AEP-K-FP in this medium showed the appearance of various peaks in the spectrum suggesting the hydrolysis of the peptide into different products, with a clear peak representing the free drug (Figure 8). The remaining detected peaks in the spectrum were likely to be identified as amino acid residues or shorter peptides sequences deriving from the hydrolysis process. This was suggested by their earlier elution due to their lower MW and reduced hydrophobicity (Figure 8). Quantitative analysis of the peaks showed that the percentage of hydrolysed FP was 69.2% out of the total amount of AEP-K-FP (Table 2).

## 4. Discussion

The transport, cellular uptake, and performance of drugs at their target sites are significant challenges in the treatment of AD and other NDs due to their limited penetration of the BBB [3]. Similar to other drugs, FP is unable to permeate through the BBB at therapeutically effective concentrations. The delivery of potential therapeutic agents, therefore, requires active-transport mechanisms to enable the transport of these molecules across the BBB. Several strategies to improve the delivery to the CNS have been developed including local injection or BBB surgical opening, facilitating the drug permeability and targeting delivery to the brain [27]. Improving permeability of drugs or other substances to cross the BBB via regulated transcytosis is deemed an attractive strategy to enable the passage of macromolecular complexes, which are not normally achieved [23].

In the recent years, brain drug delivery has focused on pathways that can exploit endogenous receptors on BBB endothelia in promoting the passage of specific particles by RMT [37]. This pathway employs the endothelial vesicular trafficking mechanism to convey molecules from blood to brain. With an appropriate receptor-targeting ligand, RMT system can shuttle numerous drugs into the brain noninvasively [50]. One of the crucial areas of concern in RMT is to find receptors that are (a) selective to the BBB vasculature and (b) highly expressed in the desired site. An additional challenge is to find ligands that are capable of taking advantage of these receptors in the transcytosis for drug delivery [23]. Upregulation of the LDL receptors at the BBB region in comparison with other endothelia supports this hypothesis via ligand recognition. Previously, studies have shown that the LDLr participated in the transport of a wide range of molecules from the blood to the brain following functionalisation of delivery systems with the binding site of the ApoE peptide [29,51].

In the present study, FP molecules were coupled to a specifically designed peptide nanocarrier, whereby the FP molecules were presented at the terminal molecular branches of a lysine added to the known AEP-mimicking peptide sequence (the ApoE 141-150 amino acids sequence) that has been shown to be able to bind LDLr [37]. A microwave-driven SPPS was used as an efficient method of synthesis. The AEP-K-FP synthesis was obtained by employing typical Fmoc-protected amino acid coupling chemistry. In order to overcome some of drawbacks associated with long chain peptides, such as intermolecular aggregation or truncated sequence [52], a capping step was introduced after each amino acid coupling. Consequently, any short chain sequences formed could be easily removed leading to a purer product, as confirmed by the MS and HPLC results of this work.

The chosen chemical modification with a lysine terminal allows a branching design of the nanocarrier that can be modified in a versatile manner depending on the targeted applications (e.g., drug and gene delivery) [53]. A previous study has shown an efficient bi-functionalisation of hyperbranched dendrimers with carboxymethyl PEG5000 and folic acid leading to high cell-specific targeting and more folic acid delivering [54]. Another study has shown that dendrimers functionalised with Tf or Lf as ligands enhanced the delivery of doxorubicin to the brain in comparison to the unfunctionalised carriers [55]. In addition the brain permeability of the opioid, peptide OX26 was potentiated when the molecule was loaded onto a functionalised dendrimer [56], while the angiopep-functionalised dendrimer has been used efficiently in gene delivery [17].

Nanocarriers’ biocompatibility is a prerequisite for its use in clinics [57]. In this work, when the AEP-K-FP was tested at concentrations up to 400 µM, it did not show any significant decrease in cellular metabolic activity in MTT assay or a significant effect on cellular membrane integrity, according to the LDH assay, after 24 and 48 h exposure periods. Hence, these results show that there is no significant cytotoxic effect on endothelial cells such as bEnd.3 and HUVEC cell lines. Cell death assessment using HPI staining also confirmed the above finding, where the cells treated with the FP-delivery system showed very low levels of apoptosis and necrosis. It can be speculated that this is due to the ability of the AEP-K-FP to cross the cells via transcytosis process without affecting their nuclei [47].

At the same time, the observation regarding the cytotoxic effect observed at higher concentrations may be attributed to the increase in permeability across the cell membrane via RMT causing excessive damage of the plasmalemma with consequent loss of the mitochondrial functions and overall cell viability [58]. Indeed, the results are consistent with results of previous studies, which demonstrated that the toxicity of ApoE peptide increases at relatively high concentrations [33].

Overall findings indicate that the noncytotoxicity of the AEP-K-FP at the concentration values used is in agreement with other studies that focussed on the effect of ApoE and ApoE-decorated nanoliposomes on brain endothelial cells up to 48 h of incubation [37]. Another study demonstrated that there was no cytotoxic effect of ApoE-modified NPs when incubated with bEnd.3 cells at different concentrations for 24 h [31].

The BEnd.3 cell line has widely been utilised as an in vitro model of drug-delivery-system penetration across the BBB, with data showing its ability to mimic sufficiently the in vivo conditions [59,60,61]. In this paper, a TEER analysis confirmed that the model was suitable to assess the transport potential of the AEP-K-FP across the BBB, as the data collected confirmed those obtained in a previous work [18]. Further validation of the model was performed to ascertain paracellular permeability. The permeability of the bEnd.3 to sucrose was significantly lower than that of HUVEC when these cells were cultured in a transwell system, the polycarbonate filter of the system showing no additional barrier effect. It was suggested that the sucrose permeability decreased in brain endothelial cell cultures that had reached confluence due to the formation of tight junctions; this was deemed an acceptable indicator of BBB model validity [62]. Another research has showed a limited sucrose permeability using the immortalised brain capillary endothelial cells [63,64].

Penetration of free FP was found to be less than 9% of its initial payload concentration, indicating levels of permeability similar to those observed in a previous study and not sufficient to induce a pharmacological effect on target cells [9]. FP permeability across the endothelial layer was improved when the drug was integrated in the carrier structure reaching a four-fold increase. It is worth highlighting that the percentage of permeability does not reflect the amount of the drug delivered to the target tissue, as each carrier bears two FP molecules that could eventually be released from delivery system upon hydrolysis of the peptide carrier. The results of the synthesised drug delivery system demonstrated its crossing of the bEnd.3 monolayer culture suggesting that it can be internalised and then undergo exocytosis following an RMT pathway. This interpretation of the data are supported by previous observations that AEP-decorated carrier systems cross the BBB model via RMT mechanism through their selective binding to the membrane LDLr [31,65].

The BBB endothelium is known for its relatively high level of LDLr expression in relation to other tissues [3,51]. This receptor mediates the transcytosis of circulating lipoproteins into the brain. ApoE is one of the known ligands for this receptor, and its amino acid sequence 141–150 is considered the recognition domain leading to the protein/receptor complex internalisation. When used to functionalize NP and liposomes, the ApoE-derived peptide facilitates the uptake of these types of carriers, and it has been proposed for applications, such as brain cancer treatment [28,66]. Likewise, through confocal microscopy and flow cytometry, our previous work has proven the increased transcytosis of AEP-modified branched nanocarrier by brain endothelial culture when compared to a nonmodified branched nanocarrier, suggesting that drugs integrated into these branched nanocarriers could indeed reach the target tissue at therapeutic doses [40].

Numerous drugs have efficiently crossed the BBB using ApoE-functionalised delivery system, such as dalargine, curcumin, and doxorubicin [28,37,65]. In this work, the AEP-functionalised delivery system has significantly enhanced the FP crossing of a brain endothelial barrier at concentrations higher than those found on a similar system based on AMT (RMT = 36%, AMT = 12%) [18], demonstrating the potential advantage in adopting RMT-driven FP delivery system in AD treatment.

In addition, the reported higher expression of the LDLr on brain endothelial is likely to increase the delivery of the required drug doses to the target tissue, thus minimizing unwanted effects of the drug on other organs and reducing the need for the administration of high and frequent doses potentially causing side effects.

FP has been found to act primarily on the PSEN (presenilin) site of the γ-secretase complex in turn prevent formation of neurotoxic Aβ42 isomer during APP cleavage [8,67,68]. The present study also evaluated FP effect on the γ-secretase enzyme activity of glial C6 cells line when coupled to the nanocarrier. Likewise, for the previously studied AMT-driven FP-functionalised nanocarrier, the results revealed that the conjugated drug still retained its action in modulatory activity of γ-secretase, albeit at a significantly lower level (*P* < 0.05). The reduced activity found could be explained by the steric hindrance of the large size molecule and/or by the spatial arrangement of the functional group which might change its reactivity. Several studies have previously supported this explanation in which drugs lacked their usual efficiency upon conjugation to carriers due to steric hindrance [69,70]. Instead, some other drugs were able to maintain their activity, while they still attach to the carriers [70].

To provide data of drug activity in the long term, the degradation of AEP-K-FP was investigated using HPLC analysis proving that the hydrolysis of the nanocarrier in conditions similar to that of the AD inflamed tissue can liberate FP molecules, thus adding the potential advantage of achieving a more protracted effect on the target tissue. Studies have reported that dendrons or peptides with such backbones are completely degraded into their building monomers, in turn releasing their payloads including the drug [71,72,73]. Furthermore, it has been shown that these biodegradable units not only help in the final dissociation of the carriers but also can reduce their toxicity through a release of lower doses by enzyme-driven proteolysis [17,74]. The peptide fragments of the carrier can be subsequently metabolised and eliminated from the body [75].

A time-course analysis of the drug liberation from the nanocarrier upon hydrolysis was not performed in this study as, albeit reduced, the drug retained its activity when bound to the nanocarrier, and it was not completely released in a free form after 24 h of incubation in the lysosomal-mimicking medium that is the recommended FP administration timescale [12]. In other words, as doses are likely to be administered to the patients daily, the improved BBB penetration and the activity of both conjugated and free FP should ensure adequate levels of administered FP at the target tissue.

The coupling of the drug to the peptide-based nanocarrier did not alter the ApoE-mimicking peptide function in promoting RMT, and predisposed the FP liberation through its grafting to the branching lysine amino groups. The latter has been demonstrated by previous studies that revealed full pharmacological action of drugs, such as methotrexate and testosterone upon degradation of the amide linkages or arms [70].

It has also been shown that dendrons or peptides can protect drugs against systemic degradation, enhancing the stability of the complex during the blood circulation [71,76,77]. It can be speculated that, as FP and other NSAIDs form complexes with plasma proteins when in free form, their complexation with the nanocarriers can also contribute to increase the amounts reaching the target tissue rather than being neutralised by their complexation with host plasma proteins [78]. In turn, this protective effect would reduce the need for relatively high doses [79].

Previous attempts to improve brain FP permeability, which were either by its entrapment in PEG-micelles [80] or by embedding FP in NP [9], did not show an increase in permeability as high as that observed in the case of the AEP-K-FP delivery system. In comparison to these studies, the present system provides the advantages of reduced molecular weight, low cytotoxicity, and long-term drug liberation—properties difficult to achieve by drug entrapping or encapsulation methods [81,82]. Together with the potential scaling-up of the synthesis of these nanocarriers, the findings of this study offer an attractive nanomedicine system for the treatment of AD by FP.

## Figures and Tables

**Figure 1 nanomaterials-10-01590-f001:**
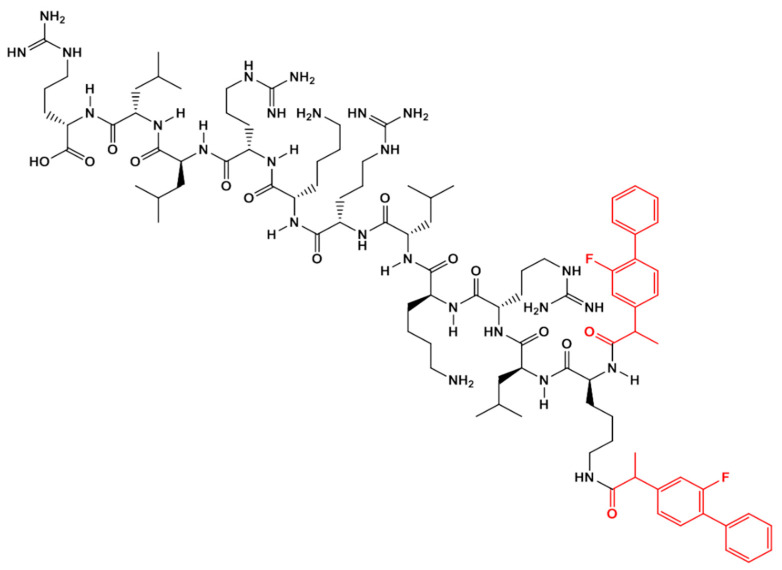
Chemical structure of the drug delivery system (AEP-K-FP) (obtained by ChemDraw Professional 15, PerkinElmer, South Carolina, US).

**Figure 2 nanomaterials-10-01590-f002:**
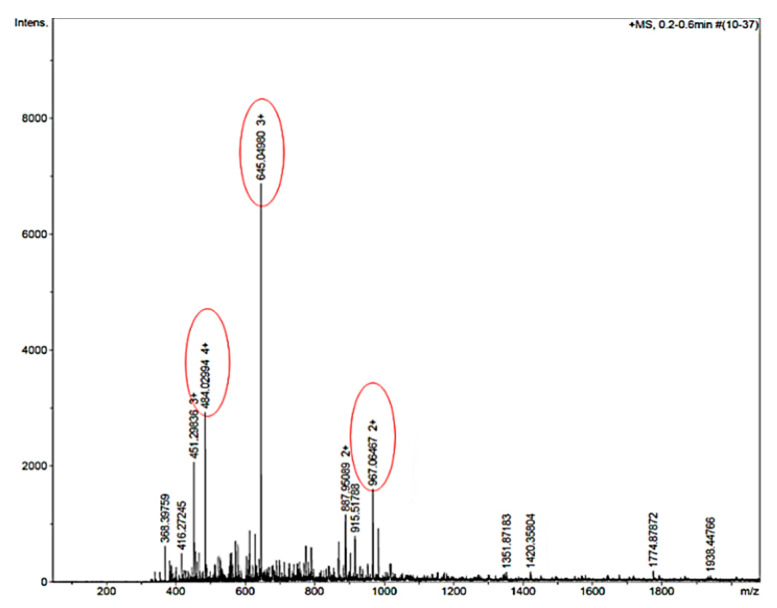
Mass spectra of AEP-K-FP shows the dominant peak (645 with triple positive charge) of intensity 6.9 × 10^3^ refers to the theoretical MW (m/z = (1932 + 1.008 × 3)/3). Alternatively, the peaks 484 with quaternary charge at 3 × 10^3^ intensity and 967 with double charge at 1.7 × 10^3^ intensity represent the related ions of AEP-K-FP (m/z = (1932 + 1.008 × 4)/4 and m/z = (1932 + 1.008 × 2)/2, respectively.

**Figure 3 nanomaterials-10-01590-f003:**
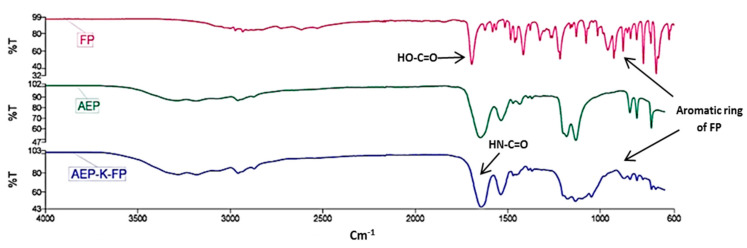
FTIR spectra of drug (FP) alone, peptide (AEP), and AEP-K-FP. The spectra show shifting peaks due to the formation of peptide linkages at 3200 and 1646 cm^−1^. The FP spectrum shows a strong carbonyl band at 1700 cm^−1^, which represents its COOH functional group that disappears in the AEP-K-FP spectrum due to amide linkage formation. In addition to the appearance of a new peak at 782 cm^−1^, it corresponds to the aromatic ring of FP with its C=C peak at 910 cm^−1^ in the AEP-K-FP spectrum, which confirms the FP attachment to the peptide.

**Figure 4 nanomaterials-10-01590-f004:**
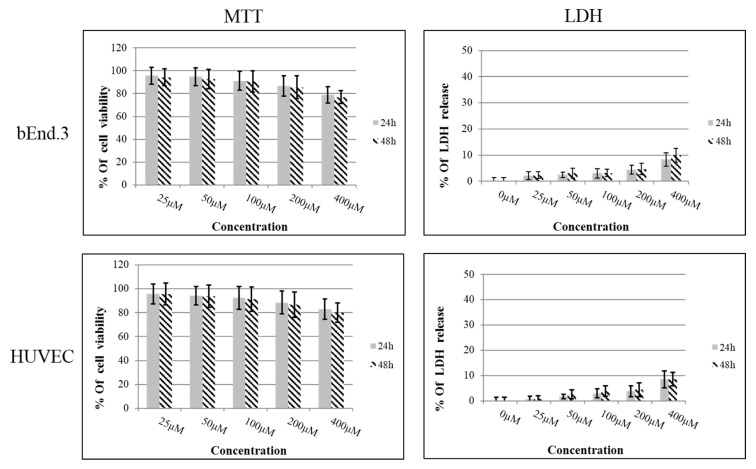
MTT and LDH results after 24 and 48 h of treatment of bEnd.3 and HUVEC AEP-K-FP. Data expressed as mean ± SD of n = 6.

**Figure 5 nanomaterials-10-01590-f005:**
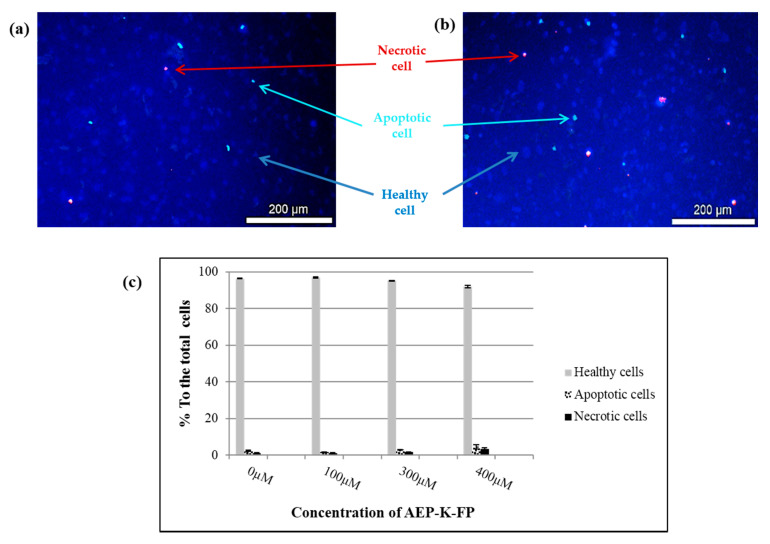
Examples of microscopic pictures after HPI staining. (**a**) Untreated cells (0 µM). (**b**) AEP-K-FP (400 µM). The blue cells represent the healthy living cells while the bright blue refers to apoptotic cells, and the red is necrotic cells. (**c**) The percentage of healthy, apoptotic, and necrotic cells to the total number of cells in each field with no significant differences from the control (*P* > 0.05). Data expressed as mean ± SD of n = 6.

**Figure 6 nanomaterials-10-01590-f006:**
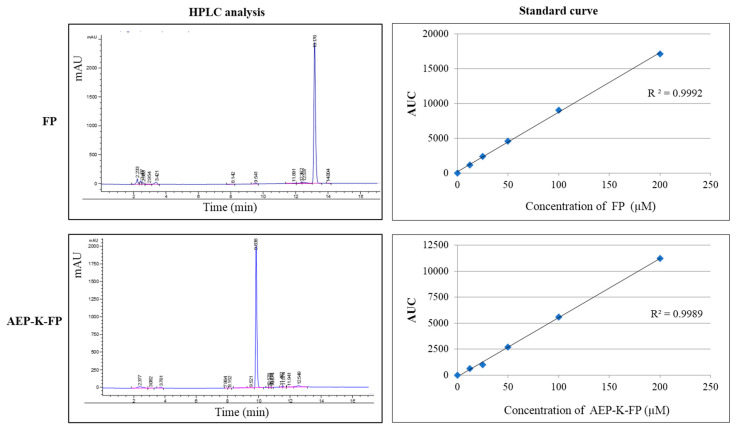
The high performance liquid chromatography (HPLC) analysis and corresponding standard curve of FP and AEP-K-FP.

**Figure 7 nanomaterials-10-01590-f007:**
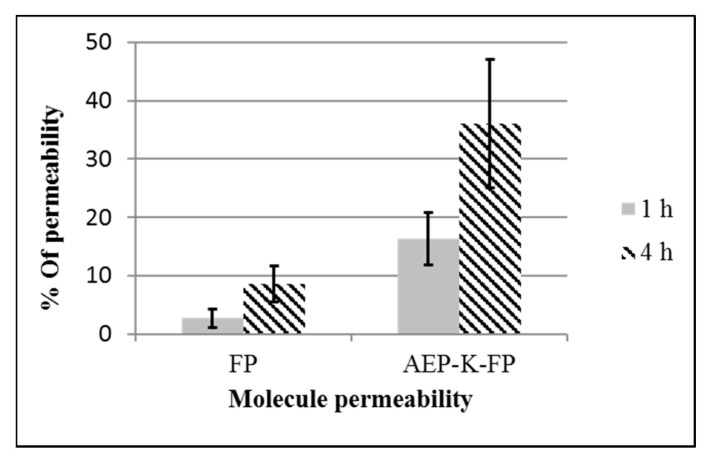
Permeability (%) of FP and AEP-K-FP across the in vitro blood brain barrier (BBB) model.

**Figure 8 nanomaterials-10-01590-f008:**
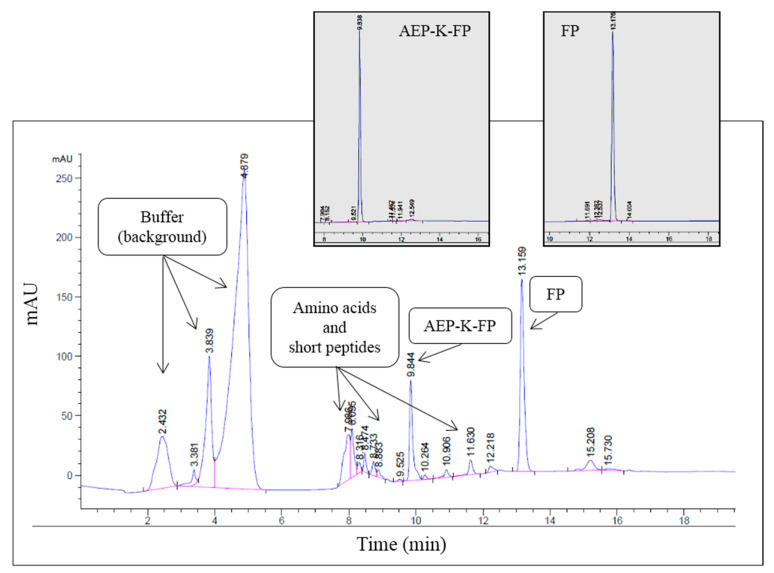
HPLC analysis of AEP-K-FP. The inserts show the elution of the initial AEP-K-FP and free FP.

**Table 1 nanomaterials-10-01590-t001:** Values of sucrose permeability.

Time	Readings	% Sucrose Permeability (Mean ± SD)		
		Membrane	HUVEC	bEnd.3
15 min	4	44.7 ± 4.1	19.6 ± 2.7	6.2 ± 1.4
30 min	4	46.8 ± 3.6	32.2 ± 3.2	9.4 ± 1.8

**Table 2 nanomaterials-10-01590-t002:** Quantification of γ-secretase enzyme after exposure to free FP and AEP-K-FP.

Factor	Readings	Mean ± SD pg/mL	*P*-Value to FP	*P*-Value to Control
Control	6	28.43 ± 8.32	< 0.05	
FP	6	8.20 ± 3.80		
AEP-K-FP	6	14.60 ± 3.51	< 0.05	< 0.05

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
