# Peer review of "A Peptide-Based Nanocarrier for an Enhanced Delivery and Targeting of Flurbiprofen into the Brain for the Treatment of Alzheimer’s Disease: An In Vitro Study"

_nanomaterials, 2020, doi:10.3390/nano10081590_

Round 1

Reviewer 1 Report

.

Author Response

The authors are grateful to the reviewer for the positive comments about the quality of our manuscript and understand the concerns expressed by him/her about the relevance to the journal and its scope.

At the same time, the authors would like to respectfully argue that the proposed carrier is not merely a branched peptide, but includes the integration of a drug in its structure that would not otherwise be able to permeate the brain blood barrier with the same efficacy. The design aimed at achieving: (i) a specifically designed peptide able to cross such a barrier, (ii) an increased drug payload through branching and (iii) a degradation of the peptide-based nanocarrier to release the free drug to the target brain cells. In addition, the paper demonstrate that when conjugated to the nanocarrier, the drug is still able to exert its pharmaceutical action on the target cells. As Nanomaterials accepts papers focussing on the design and synthesis of a wide range of nanomaterials (including for example quantum dots) and their applications, we would like to point out that our work is the combination of an application-driven design, synthesis and characterisation and in vitro proof of concept of the above listed properties.

The authors agree with the reviewer that the used terminology poly(epsilon-Lysine) is in this instance wrong. We erroneously used it as in the past we have published papers where multiple branching generations were synthesised and the term has become entrenched in the communication and writing of our team without sufficient consideration for the actual characteristics of the macromolecule here studied. We apologise for that and have amended the terminology throughout the manuscript.

All the other amendments recommended by the reviewer have now been made in the revised manuscript. In particular, the paper title has been changed and the English throughout the manuscript have been improved. We prefer to keep the FTIR as this show the successful coupling of the drug molecules. 

Reviewer 2 Report

The article presents in a scientific and appropriate manner a new approach regarding the delivery of potential therapeutic drugs across the BBB. Very interesting and innovative protocol and quite promising results!

Author Response

The authors are grateful to the reviewer for the positive comments about our manuscript.

Reviewer 3 Report

Overall, this is a nicely written work and contains important information for delivering and targeting molecule into the brain across the BBB, despite this in vitro.  I have two main issues with it however, both related to γ-secretase, and those are: the rate of the hydrolysis of AEP-K-FP and consequently the pharmacological effect on the γ-secretase enzyme activity.

In crossing the BBB, AEP-K-FP showed greater permeability (36%) than free FP (8.5%), and no one cytotoxicity as FP. When cells in culture are treated for 4h with AEP-K-FP and free FP, the quantification of γ-secretase activity showed that free FP had a stronger pharmacological effect on the enzyme inhibition. The question is: Is this effect due to the fact that free FP is immediately available for cells while the AEP-K-FP hydrolysis process could take a long time to release the free FP? What is the time and rate of hydrolysis? Is all conjugated FP released over time?

Maybe a time course analysis could be useful not only to better understand that observed difference in the y-secretase inhibition (~2 lower than free FP) but also to evaluate the potential advantage of conjugated FP in achieving a more protracted effect on the target tissue.

Minor points:

a)In the Results, under Cytotoxicity and LDH assay: While describing Fig 5a,b, the authors note no cytoxicity in the concentration range used, but the authors need to specify the concentration used in Figure 5b. In addition, it may be useful to insert clearer HPI stained images of all the concentrations (0,100,300,400)

b)In the Results, at row 320: “…..more than 6 times (16.3%) after 1 h, and 4 times (35.9%) after 4 h with significant differences (P<0.001) from that of free FP levels…” the authors need to specify the free FP levels even after 1h.

c)In the Results, at row 344: “The decrease was also significant (P<0.05) in comparison to the levels of the free drug (Table 2)” the authors need to clarify this sentence. If they refer to the average value of 14.6(AEP-K-FP) respect to 8.2 (FP), obviously not seems a reduction.

d)In the Results, under degradation analysis and drug release: While describing Fig 8, the authors note a clear peak representing the free drug but Fig 8 also show a clear peak of AEP-K-FP. Would it be possible to quantify the percentage of hydrolyzed out of the total amount?

Author Response

The authors are grateful to the reviewer for the positive comments and for the most valuable advice given to improve the quality of the manuscript.

We agree with the reviewer that a 'time-course analysis' of the drug liberation from the nanocarrier would have provided more information about its effect on the target cells. The reason why a more accurate study was not performed is that the hydrolysis studies show that the drug retained its activity (albeit reduced) when bound to the nanocarrier and it was not completely released in a free form after 24 h of incubation in a lysosomal-mimicking medium. As in therapy doses are likely to be administered to the patients daily, the improved BBB penetration and the activity of both conjugated and free FP should ensure adequate levels of administered FP at the target tissue. This comments have now been included as part of the Discussion section.

We also agree with the reviewer that the different efficacy observed in the coupled and free drug may be due to its different availability to the target cells when bound or free. All these considerations have now been included in the Discussion of the revised manuscript.

As far as the Minor Points are concerned, they are addressed as it follows:

  1. the authors need to specify the concentration used in Figure 5b .  Concentration now Specified

 In addition, it may be useful to insert clearer HPI stained images of all the concentrations (0,100,300,400)

We respectfully argue that adding additional images will not add any further information and make the all figure panel difficult to visualise. Therefore, we would like to ask the reviewer and the editor to kindly accept the image in its current format.

b) specify the free FP levels even after 1h. done (line 327)

c)In the Results, at row 344: “The decrease was also significant (P<0.05) in comparison to the levels of the free drug (Table 2)” the authors need to clarify this sentence

 changed ( line 341)

d) While describing Fig 8, the authors note a clear peak representing the free drug but Fig 8 also show a clear peak of AEP-K-FP. Would it be possible to quantify the percentage of hydrolyzed out of the total amount?

 included. (354)

Reviewer 4 Report

The contribution by Al-azzawi et al. describes a novel dendron to allow passage of the drug Flurbiprofen across the blood-brain barrier. This is an interesting area because it could open a new pathway for treating Alzheimer's disease. The work has been carefully carried out, and the conclusions are in general supported by the experiments.

The manuscript could be improved in a number of ways:

a) The main problem is the poor quality of the micrographs in Fig. 5. This reviewer was unable to distinguish between healthy, apoptotic and necrotic cells in the pictures. They must be improved.

b) Permeability is expressed in Fig. 7 as "Permeability %". But some absolute figures are required, such as "mol of substrate/square centimeter of cell barrier x minute", or the like.

c) Gamma-secretase activity (Table 2) is expressed as pg/mL. These are not approved units for enzyme activity. Instead, mol of substrate per unit time should be used. (Or per unit time x mass of protein).

Moreover, there are many typos throughout the manuscript. To give but a few examples, (line 21) dendrons/dendron, (line 27) functinalised/functionalised, (line 138) triisoprpylsilane/triisopropylsilane, and many others. Careful edition is required.

There are several instances of paragraphs consisting of a single sentence. Paragraphs should contain more than one sentence.

Author Response

The authors are grateful to the reviewer for the feedback and suggestions made to improve the manuscript. All issues have been addressed in the revised version of the manuscript.

In particular, the following amendments have been made

a) The main problem is the poor quality of the micrographs in Fig. 5. This reviewer was unable to distinguish between healthy, apoptotic and necrotic cells in the pictures. They must be improved. Image quality has been improved

b) Permeability is expressed in Fig. 7 as "Permeability %". But some absolute figures are required, such as "mol of substrate/square centimeter of cell barrier x minute", or the like.  We respectfully argue that it is more reliable to express certain parameters of drugs such as bioavailability and permeability as % rather than other units, since % gives better indication of the overall results enabling the comparison with other models published in literature. In particular, using moles would not be adequate because of the different molecular weight and drug payload (2 molecules of drug) of the nanocarrier when compared to the free drug.

Gamma-secretase activity (Table 2) is expressed as pg/mL. These are not approved units for enzyme activity. Instead, mol of substrate per unit time should be used. (Or per unit time x mass of protein). The data were expressed according to the instructions provided with the assay kit and its relative standard curve. There was no sufficient information in the kit to convert the data into units such as the international unit. Therefore, we would like to keep the data as in pg/mL.

d) there are many typos throughout the manuscript. The English of the manuscript has been reviewed